# Spin-Selected Dual-Wavelength Plasmonic Metalenses

**DOI:** 10.3390/nano9050761

**Published:** 2019-05-17

**Authors:** Wei Wang, Zehan Zhao, Chong Guo, Kai Guo, Zhongyi Guo

**Affiliations:** 1Department of Mathematics and Physics, Shijiazhuang Tiedao University, Shijiazhuang 050043, China; xvxvxiaozhao@163.com (Z.Z.); guocong122@163.com (C.G.); 2School of Computer and Information, Hefei University of Technology, Hefei 230009, China; kai.guo@hfut.edu.cn

**Keywords:** metasurface, spin, phase modulation, metalens

## Abstract

Several novel spin-selected dual-wavelength metalenses have been proposed and investigated based on the plasmonic metasurface consisting of two kinds of rotary rectangle gap nanoantennas (RGN), which are designed based on merging two or four polarity-inverse lenses corresponding to different wavelengths (765 and 1300 nm). The spin-selected dual-wavelength metalenses with two similar and two different vertical and lateral focal points have also been proposed respectively, which can focus two wavelengths with inverse spin states to arbitrary special positions. The three-dimensional metalens with four focal points have also been proposed, which can focus four beams with inverse spin states and different wavelengths to preset positions. Moreover, a spin-dependent achromatic metalens has also been proposed, which can focus left circularly polarized (LCP) incidence with different wavelengths to the same position. Our work opens up new avenues toward establishing novel spin-selected and wavelength-selected metadevices, and is significant for the development of spin-controlled photonics and particles manipulation. In addition, it provides a new idea for solving the problem of data transmission from optical fiber communication to visible light communication.

## 1. Introduction

With the developments in nanofabrication technology, metasurfaces consisting of nanoantennas have attracted great research interests, which can tailor the amplitudes, phase, and polarization states of the incident electromagnetic waves [1,2,3,4]. Rather than relying on gradual phase accumulation through light propagation, every nanoantenna of the metasurface achieves abrupt phase discontinuity at the interfaces. Through adjusting the shape and geometry parameters of the optical antennas, the full 2π phase modulation can be achieved, which provide an alternative approach for the wave control and manipulations. Recently, many planar optical components based on plasmonic metamaterials and metasurfaces have been successfully designed and realized, such as vortex phase plates [5,6,7], lenses [8,9,10,11], wave plates [12,13,14,15], holograms [16,17,18], nonlinear harmonic signal generator [19], plasmonic biosensors [20,21], and so on.

Furthermore, two or more wavelength multiplexing metasurfaces have gained considerable attentions in recent years [22,23,24,25,26,27,28,29]. They are free from the chromatic aberration and beneficial to the applications of metasurfaces in broadband and multispectral devices. Ding et al. [30] have proposed a dual-wavelength metalens in the terahertz range, which is only suitable for x-polarized light. Li et al. [31] have designed several dual-wavelength metalenses in the visible range, which is only suitable for left circularly polarized (LCP) light. Moreover, recent works [32,33,34,35,36] show that the chromatic effect is eliminated over a continuous wavelength range via a single metasurface design, and are also only suitable for one kind of polarized light. In these papers, the multi-wavelength metalenses have been realized only for the linearly polarized or circularly polarized (CP) light. Most importantly, the reported multi-wavelength metalenses for CP light will only achieve LCP or right circularly polarized (RCP) focusing [22,23], without focusing LCP and RCP incidences to different positions simultaneously. On the other hand, spin-selected metasurfaces for a particular wavelength have also been proposed in recent years [37,38,39,40,41,42]. They can not only separate photons with different spin states, but also realize the manipulation to different spin light, and the manipulated spin states can be focused to preset positions. It is worth mentioning that the reported spin-selected metasurfaces only work at a single determined wavelength. In the previous papers, the multi-wavelength metalenses only worked for a kind of polarized light [22,23], or possessed polarization-independent properties [43], and the spin-selected metalens only worked at a single wavelength [37,38,39,40,41,42]. However, as far as we know, the metalenses, which possess spin-selected and wavelength-selected function simultaneously, is yet to be explored. Meanwhile, the metalens, which can focus four beams including LCP and RCP light with different wavelengths to different positions simultaneously, is scarcely reported. Designing metasurfaces to realize a polarization and wavelength analyzer is of great significance to the development of metasurface devices.

In this paper, we propose several spin-related dual-wavelength metalenses, which are constructed by two kinds of rotary rectangle gap nanoantennas (RGN) corresponding to different wavelengths. To meet near-infrared optical fiber communication and future visible light communication, 765 nm and 1300 nm are selected as target wavelengths. Three kinds of spin-selected dual-wavelength metalenses are designed, in which different spin photons with different wavelengths can be focused to same or different positions respectively. A three-dimensional spin-selected dual-wavelength metalens is designed, four beams with two spin states and two wavelengths can be focused to the preset positions. One spin-dependent achromatic metalens is also designed, in which the LCP incident light with different wavelength can be focused to the same positions. These spin-selected and wavelength-selected metalenses are meaningful for the development of particles manipulation and detection techniques.

## 2. Design and Methods

As shown in Figure 1a, the RGNs with spatial variant orientation angles of φ are designed as the basic unit cell. For avoiding coupling effects between neighboring RGNs, the unit cell period of P is set as 400 nm. When CP light is normally incident from the side of glass substrate, the transmitted light typically consists of two parts: one is co-polarized light without any phase delay, and another is cross-polarized light with an additional phase delay of ±2φ, where “±” depends on the incident spin states. The additional phase delay, known as Pancharatnam–Berry (PB) phase, is independent of the incident wavelength, but the transmission of cross-polarized light strongly depends on the RGNs’ resonance through changing the length and width. Based on the theory of PB phase, the spin-selected metalens can be designed through multiplexing two opposite polarity metalenses. Based on the resonance theory, the RGNs with different lengths and widths will possess different transmissions, so the dual-wavelength metalens can be designed through multiplexing two metalenses for two incident wavelengths. With the multiplexing idea, the spin-selected and wavelength-selected metalens can be designed through multiplexing two or several-opposite-polarity metalenses at two incident wavelengths.

Based on the idea of multiplexing, a dual-wavelength metalens can be achieved through arranging the two types of RGNs properly. To realize the wavelength-selective focusing, research on the resonance of the RGNs is crucial. Here, the finite element method is used to perform the numerical simulations. We set the height of the gold and glass substrate as 100 and 200 nm, respectively. Periodic boundary conditions are set along the x- and y-directions, and perfectly matched layers are given in the z-direction. Figure 1b shows the influence of geometric parameters (L and W) on the resonance characteristics of the RGNs, in which both the resonant wavelength and transmission peak value increase along with increasing RGN length, but the resonant wavelength decreases and transmission peak value increases along with increasing RGN width. Through further optimizations, the optimized geometric parameters for the wavelengths of 765 nm and 1300 nm are selected as $W_{1}=150\;\mathrm{nm},\;L_{1}=200\;\mathrm{nm}$ and $W_{2}=50\;\mathrm{nm},\;L_{2}=300\;\mathrm{nm}$, respectively. The normalized transmissions for two types of RGNs in the range from 700 nm to 1500 nm are displayed in Figure 1c. Their resonance peaks are located at two wavelengths of 765 nm (A), and 1300 nm (B). More importantly, both the transmissions of RGN A at 1300 nm and RGN B at 765 nm are relatively lower. Therefore, when incident wavelength is 1300 nm or 765 nm, only one kind of RGN is predominantly activated to induce the geometric PB phase for the dual-wavelength metasurface composed of two kinds of RGNs. The cross-talks between them are weak and can be neglected, and the transmitted light could be quite pure. Therefore, the expectant wavefront (765 nm/1300 nm) can be independently manipulated by changing the RGNs’ (A/B) orientations.

For designing an ordinary planar metalens focusing at the X–Z plane, according to the Fermat’s principle, the phase distributions of the metalens along *x*-axis can be expressed as follows:(1)$$\Phi \left(x\right)=\pm \frac{2\pi }{\lambda }\left(\sqrt{x^{2}+f^{2}}-f\right)\;$$
where *f* is the focal length, *λ* is the incident wavelength, “+” corresponds to LCP incidence, and “−” corresponds to RCP incidence. In fact, the dual-wavelength metalens can be realized by alternately arranging two kinds of rotating RGNs A and B along *x* axis. The spin-selected metalens can be made up by two metalenses with inverse polarity, and the phase shift possess opposite signs for two polarity-inverse metalenses. Therefore, the spin-selected and wavelength-selected metalens can be made up by two polarity-inverse metalenses, which correspond to different incident wavelengths. Figure 2 describes the schematic diagram of the spin-selected dual-wavelength metalens, which can achieve any focusing for LCP and RCP incident light with different wavelengths. According to different focusing locations and the spin states of incident light, four spin-selected and an achromatic dual-wavelength metalenses can be realized.

## 3. Results and Discussions

We have designed a spin-selected dual-wavelength metalens with the same focusing length, and it can focus the LCP light and RCP light with different wavelengths of 765 nm and 1300 nm simultaneously. The metalens with a focal length of 8 μm can be realized through alternately arranging two kinds of RGNs. The phase distributions of the metalens along *x*-axis can be expressed as follows:(2)$$\Phi \left(x\right)=\left\{\begin{array}{c} \frac{2\pi }{\lambda_{1L}}\left(\sqrt{x^{2}+f^{2}}-f\right)\;,x=2nP\\ -\frac{2\pi }{\lambda_{2R}}\left(\sqrt{x^{2}+f^{2}}-f\right)\;,x=\left(2n+1\right)P \end{array}\right.$$
where $\lambda_{1L}$ is the first wavelength of incident LCP light, $\lambda_{2R}$ is the second wavelength of incident RCP light. The concrete calculated phase distributions for the wavelengths of 765 nm and 1300 nm have been shown in Figure 3a. The metalens is composed of 49 resonators, as shown in Figure 3b, and the numerical aperture can be calculated as NA = 0.77, which is quite high. The focusing property of the spin-selected dual-wavelength metalens is simulated with the finite element method. In the numerical calculation, perfectly matched layer boundary conditions were used in the z-direction while periodic boundary conditions were applied in the y-direction. For LCP incidence with the wavelength of 765 nm, the transmitted RCP light is focused perfectly at the pre-setting location, as shown in Figure 3c, which is 8 μm behind the metalens. For RCP incidence with the wavelength of 1300 nm, Figure 3d shows that the transmitted LCP light is also focused at 8 μm behind the metalens. Therefore, the designed metalens is a spin-selected dual-wavelength metalens with the same focal length of 8 μm. The efficiency of the metalens is 7.0% and 12.7% for the incident wavelength of 1300 nm and 765 nm, respectively. As demonstrated in Figure 3c,d, for two wavelengths with inverse spin-states, the full widths at half maximum (FWHMs) are 391 nm and 577 nm, respectively. The designed metalens possesses relatively high focusing performances for both wavelengths. Because two foci possess inverse spin states, in theory, the designed metalens can be used to trap and turn particles along clockwise or anticlockwise directions by manipulating the incident wavelengths.

We have also designed two spin-selected dual-wavelength metalenses with the focus displacements along vertical and lateral directions, respectively. The phase distributions of the metalens with two longitudinal focuses along *x*-axis can be expressed as follows:(3)$$\Phi \left(x\right)=\left\{\begin{array}{c} \frac{2\pi }{\lambda_{1L}}\left(\sqrt{x^{2}+f_{L}^{2}}-f_{L}\right)\;,x=2nP\\ -\frac{2\pi }{\lambda_{2R}}\left(\sqrt{x^{2}+f_{R}{}^{2}}-f_{R}\right)\;,x=\left(2n+1\right)P \end{array}\right.$$
where $\lambda_{1L}$, $f_{L}$ are the first wavelength and focal length corresponding to LCP incidence, $\lambda_{2R}$, $f_{R}$ are the second wavelength and focal length corresponding to RCP incidence, and n is an integer. We first design a spin-selected dual-wavelength metalens with the focus displacements along the vertical direction, in which the focal length is 8 μm for the LCP incidence of 765 nm, and the focal length is 4 μm for the RCP incidence of 1300 nm. Figure 4a,b shows the metalens can well focus the light to the preset positions for two incident wavelengths with inverse spin states, respectively. The FWHMs are 392 nm and 485 nm for LCP (765 nm) and RCP (1300 nm) incident light, respectively. The NA = 0.77 and 0.93 for LCP (765 nm) and RCP (1300 nm) incident light, respectively. The designed metalens with two longitudinal focuses possesses relatively high focusing performances for both wavelengths. The efficiency of the metalens is 12.4% and 8.2% for the incident wavelength of 765 nm and 1300 nm, respectively. The phase distributions of the metalens with two lateral focuses along *x*-axis can be expressed as follows:(4)$$\Phi \left(x\right)=\left\{\begin{array}{c} \frac{2\pi }{\lambda_{1L}}\left(\sqrt{{\left(x-x_{0}\right)}^{2}+f^{2}}-f\right),\;x=2nP\\ -\frac{2\pi }{\lambda_{2R}}\left(\sqrt{{\left(x+x_{0}\right)}^{2}+f^{2}}-f\right),\;x=\left(2n+1\right)P \end{array}\right.$$


The spin-selected dual-wavelength metalens with the focus displacements along lateral directions can be designed. For the designed spin-selected dual-wavelength metalens with two lateral focuses along x-directions, Figure 4c,d shows intensity distributions of the transmitted RCP and LCP lights under the LCP incidence of 765 nm and RCP incidence of 1300 nm respectively. Here the transmitted RCP light is focused to the position of 2.2 μm, and 8 μm with the LCP incidence of 765 nm, and the transmitted LCP light is focused to the position of −2.2 μm, and 8 μm with the RCP incidence of 1300 nm. The FWHMs are 385 nm and 559 nm for LCP (765 nm) and RCP (1300 nm) incident light, respectively. NA = 0.77 for LCP (765 nm) and RCP (1300 nm) incident light. The designed metalens with two lateral focuses possesses relatively high focusing performances for both wavelengths. The efficiency of the metalens is 8.3% and 7.1% for the incident wavelength of 765 nm and 1300 nm, respectively. Both metalenses are able to separate two incident lights completely. Thus, the designed spin-selected dual-wavelength metalenses can achieve focusing effects at the arbitrary positions for inverse spin states with the corresponding wavelengths.

In order to testify the superiority of our designs, we try to design a 3D spin-selected dual-wavelength metalens, which can focus four incident beams with LCP and RCP in 765 and 1300 nm simultaneously. Through setting the focal positions as $\left(x_{0},0\right),\left(-x_{0},0\right),\left(0,y_{0}\right),\left(0,-y_{0}\right)$ for the LCP (1300 nm), RCP (1300 nm), LCP (765 nm), RCP (765 nm) incidences respectively, where $x_{0}$ = $y_{0}$ = 1.8 μm, the required phase distributions for four incidences can be obtained based on the following equations:(5)$$\Phi \left(x\right)=\left\{\begin{array}{c} 2\pi /\lambda_{1L}\left(\sqrt{{\left(x-x_{0}\right)}^{2}+y^{2}+f^{2}}-f\right)\\ -2\pi /\lambda_{1R}\left(\sqrt{{\left(x+x_{0}\right)}^{2}+y^{2}+f^{2}}-f\right)\\ 2\pi /\lambda_{2L}\left(\sqrt{x^{2}+{\left(y-y_{0}\right)}^{2}+f^{2}}-f\right)\\ -2\pi /\lambda_{2R}\left(\sqrt{x^{2}+{\left(y+y_{0}\right)}^{2}+f^{2}}-f\right) \end{array}\right.$$
where $\lambda_{1}$ = 1300 nm, $\lambda_{2}$ = 765 nm, and *f* = 4 μm. The designed 3D spin-selected dual-wavelength metalens is shown in Figure 5, where the supercell is composed of four kinds of RGNs that are used to focus four kinds of incidences with different spin states and different wavelengths. The intensity distributions at the focal plane are shown in Figure 6a–d under the incidences of the LCP with the wavelength of 765 nm (Figure 6a), RCP with the wavelength of 765 nm (Figure 6b), LCP with the wavelength of 1300 nm (Figure 6c), and RCP with the wavelength of 1300 nm (Figure 6d)), respectively. As expected, the light is focused very well on the preset position, in which the LCP and RCP incidences with the same wavelength are focused at the symmetrical positions. Therefore, our designed 3D metalens possess spin-selection and wavelength-selection characteristics simultaneously. Figure 6e,f shows the bifocal focusing phenomenon under X-linearly polarized (XLP) incidences with the wavelengths of 765 nm and 1300 nm respectively. Here, there are two foci possessing opposite spin states because XLP light can be considered as the superposition of equal-intensity LCP and RCP lights. However, dual-focusing results show reduced focusing quality and uneven energy distribution, which may originate from the interaction between adjacent unit cells. Meanwhile, our designed metalens’ scale is only 6.4 μm × 6.4 μm, and if we increase the size of metalens, the focusing effect will be better. And we can find that the diameter of the focus in 765 nm is much smaller, which may be due to the better focusing effect for shorter wavelength. Therefore, our designed 3D metalens can be served as a linear/circular polarization and wavelength analyzer simultaneously.

We finally designed an achromatic metalens for focusing LCP lights under 765 nm and 1300 nm simultaneously. Under the LCP incidence with the wavelength of 765 nm, the focusing phenomenon is shown in Figure 7a. The transmitted RCP light is nicely focused at the preset location that is 8 μm behind the metalens. Under the LCP incidence with the wavelength of 1300 nm, Figure 7b shows that the transmitted RCP light is also nicely focused at 8 μm behind the metalens. The FWHMs are 300 nm and 316 nm for LCP incident light with different wavelengths. Therefore, the designed metalens is a dual-wavelength metalens with a focal length of 8 μm, which can be considered as an achromatic metalens for the wavelengths of 765 nm and 1300 nm.

## 4. Conclusions

In conclusion, we have demonstrated several spin-selected dual-wavelength metalenses, which can be used to modulate the focusing performances of different incident wavelengths with inverse spin states. Through spatial interleaving two polarity-inverse metalenses that are designed for two different wavelengths, we can achieve spin-selected metalens manipulating at two wavelengths simultaneously. We respectively designed three spin-selected dual-wavelength metalenses, which can focus two wavelengths with inverse spin states on arbitrary positions. We designed a 3D spin-selected dual-wavelength metalens, which can focus four incident spin states at two wavelengths to preset positions simultaneously, which can be served as a linear/circular polarization and wavelength analyzer. We have also designed an achromatic metalens under LCP incidence for the wavelengths of 765 nm and 1300 nm. The research of spin-selected and dual- wavelength metalens paves the way to future applications in advanced imaging, color display technologies, and spin-based photonics devices.

## Figures and Tables

**Figure 1 nanomaterials-09-00761-f001:**
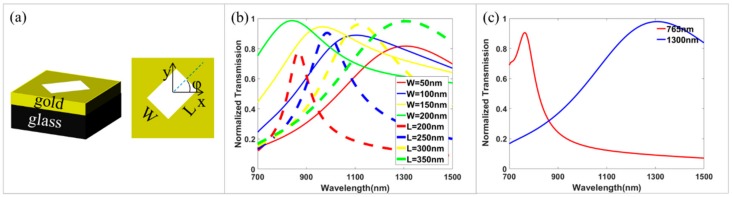
(**a**) 3D view and top view of basic unit cells. (**b**) Transmission of the rectangle gap nanoantennas (RGN) (with different L and W) as a function of the wavelength. (**c**) The normalized transmission of two RGNs under normal left circularly polarized (LCP) incidence light with the wavelength from 700 nm to 1500 nm.

**Figure 2 nanomaterials-09-00761-f002:**
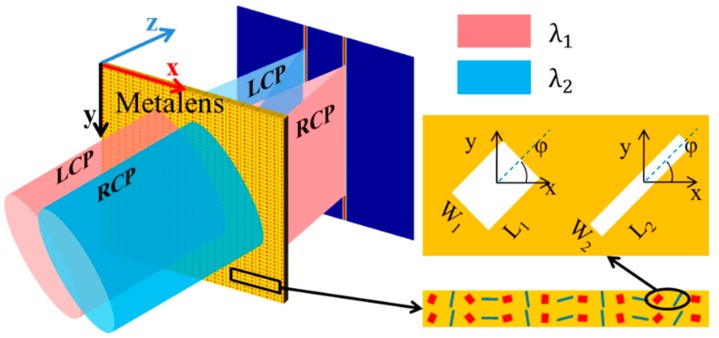
Schematic of the spin-selected dual-wavelength metalens for focusing different circularly polarized light with different wavelengths.

**Figure 3 nanomaterials-09-00761-f003:**
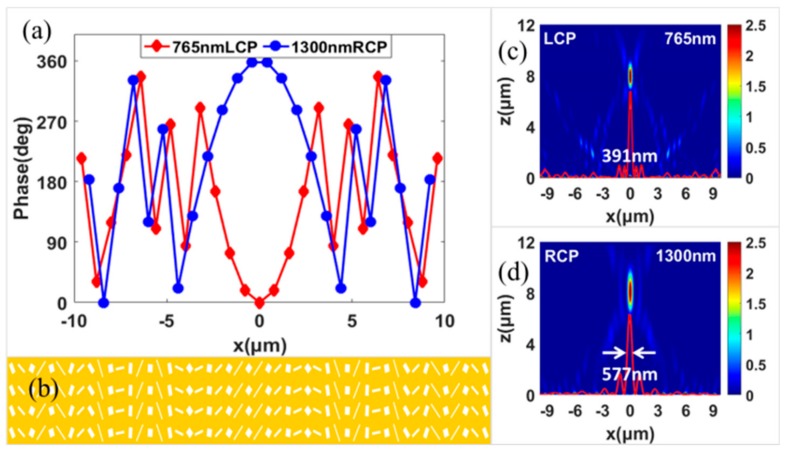
Phase shift distributions along *x*-axis (**a**) and top view (**b**) of the spin-selected dual-wavelength metalens with the same focus. Simulated intensity (${\left|E\right|}^{2}$) distributions in the X–Z plane of spin-selected dual-wavelength metalens under the LCP of 765 nm (**c**), RCP of 1300 nm (**d**) illuminations. The red curves show the concrete profiles of the full widths at half maximums (FWHMs) of the focus.

**Figure 4 nanomaterials-09-00761-f004:**
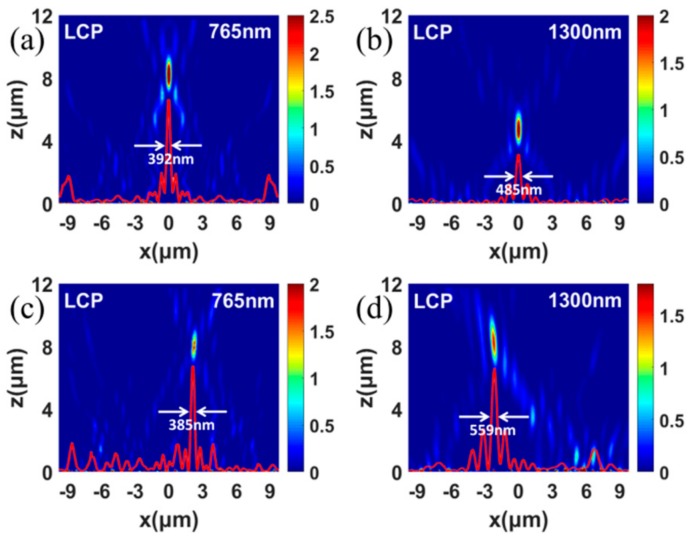
Simulated intensity distributions in the X–Z plane of the designed spin-selected dual-wavelength metalens with the focus displacement along vertical direction under the LCP incidence of 765 nm (**a**), RCP incidence of 1300 nm (**b**). Simulated intensity distributions in the X–Z plane of spin-selected dual-wavelength metalens with the focuses displacement along lateral direction under the LCP incidence of 765 nm (**c**), RCP incidence of 1300 nm (**d**). The red curves show the concrete profiles of the FWHMs of the focus.

**Figure 5 nanomaterials-09-00761-f005:**
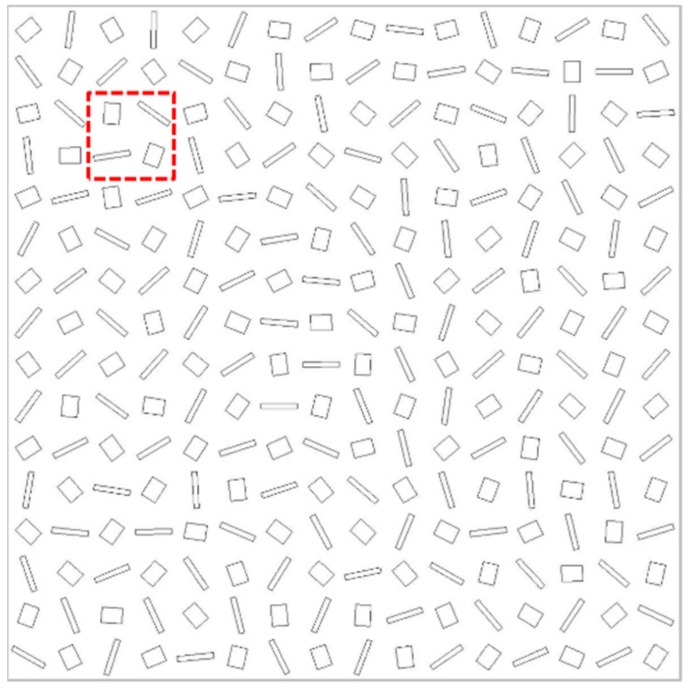
Schematic of 3D metalens for focusing different circularly polarized lights with different wavelengths.

**Figure 6 nanomaterials-09-00761-f006:**
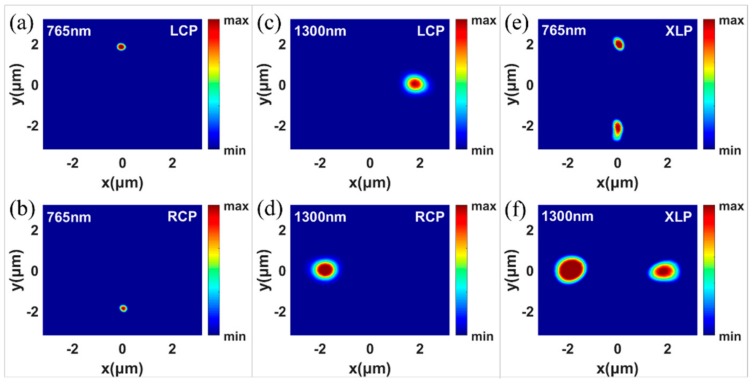
Simulated intensity distributions in the X–Y plane of 3D metalens with four focuses under the LCP of 765 nm (**a**), RCP of 765 nm (**b**), XLP of 765 nm (**c**), LCP of 1300 nm (**d**), RCP of 1300 nm (**e**) and, XLP of 1300 nm (**f**) illuminations.

**Figure 7 nanomaterials-09-00761-f007:**
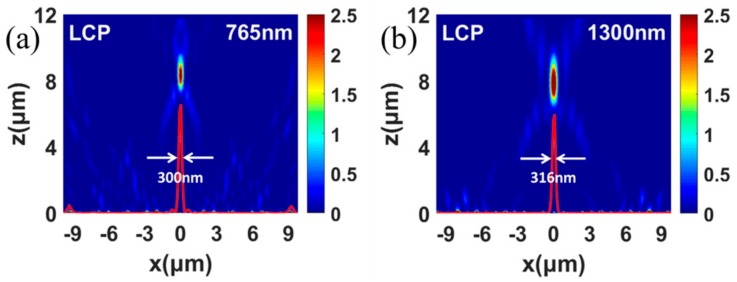
Simulated intensity distributions in the X–Z plane of achromatic metalens under the LCP incidence with the wavelength of 765 nm (**a**), and 1300 nm (**b**). The red curves show the concrete profiles of the FWHMs of the focus.

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
