# Peer review of "Spin-Selected Dual-Wavelength Plasmonic Metalenses"

_nanomaterials, 2019, doi:10.3390/nano9050761_

Round 1

Reviewer 1 Report

1. The novelties of the paper are not so clear in the following sense: Although the authors refer to them in the Introduction, this is not so prominent in the text. For example, in which manner the proposed structure is superior to other existing implementations?

2. The analysis of the results and the cases examined are rather limited. Thus please try to include comparisons with existing schemes and more informative plots (apart from surface diagrams) in order to convince about the efficiency of the metalens.

Author Response

See the Attached files.

Reviewer 2 Report

In the submitted manuscript, the authors numerically analyzed and reported the spectral response of engineered spin-selected dual-wavelength metalenses with two similar vertical and lateral different focal points. The researchers here developed 3D spin-selected dual-wavelength metalens, enable to focus four incident spin states at two wavelengths to preset positions simultaneously, which can be served as a linear/circular polarization and wavelength analyzer. It is shown that the designed achromatic metalenses under LCP incidence for the wavelengths of 765 nm and 1300 nm. Although the work looks interesting, it needs for some important revisions and clarifications that are listed below. I do suggest the authors to addressed the comments and provide a revised version of manuscript.

1) The writing quality must be enhanced and it needs for a serious polish.

2) In the bibliography section of the work, and at the end of first paragraph: Provide much more examples of the use of plasmonic metamaterials and metasurfaces in recent works such as nonlinear harmonic signal generation and biosensing applications (i.e. Nano Letters 2019, 19 (1), 605–611, ACS Photonics 2016, 3, 2308-2314, ACS Sensors 2017, 2 (9), 1359-1368).

3) In the Methods section, the type of FEM analysis has not been reported. The authors should provide the commercial package that they used and cite it with full numerical settings details. 

4) The E-field maps in the entire work needs for the clarification. It should be described that f the provided snapshots are E-field enhancement (|E|^2)? or something else?

5) Given that plasmonic structure are lossy, I am curious to know that how the structure is efficient? What is the influence of possible attention factors on the spin-selected dual-wavelength metalens?

6) In Figure 1b and 1c: The transmission spectra is fairly sharp for the ~780 nm, however for ~1300 nm, the lineshape is broad. How this difference happened? and also how this broadness affects the operation performance of metalens?

Author Response

See the Attached files.

Reviewer 3 Report

This manuscript presents numerical studies of light propagation through perforated film with periodic array of holes of special shapes. This device is called by the authors as a “spin-selected dual-wavelength plasmonic metalenses”. Two light beams of wavelengths lambda_1 and lambda_2 and respectively with right-circular and left-circular polarizations are impinged on the metal surface. Then the authors analyzed the intensity and focusing ability of the transmitted light in X-Z plane.

This is absolutely unreadable paper. It requires significant and  major revision. The authors should explain in details the physical bases of the discussed phenomena and present detail mathematical description as well as clear pictures.

Technical Comments:

1)      In text under Fig. 2 it is not clear where are x,y, and z axes and directions. Expression for Phi(x) in text after Fig. 2 should be explained and given separately with numbering but not inside the text. Similarly in respect to other mathematical evaluations. A proper picture should be presented.

2)      What the role of plasmons which are mentioned in the title of the manuscript.

3)      The English should be improved. Phrases like “generally speaking” in page 2, “that is to say’ in page 3 etc are not scientific style

Author Response

See the Attached files.

Round 2

Reviewer 1 Report

The authors have answered to the majority of my comments.

Reviewer 2 Report

Publishable as is.

Reviewer 3 Report

I have already reviewed this manuscript. The authors took into account some of my comments and I can recommend the manuscript for publication.